# Psychometric properties of the Brazilian-Portuguese Flow State Scale Short (FSS-BR-S)

Ig Ibert Bittencourt[1]*, Leogildo Freires[2], Yu Lu[3]*, Geiser Chalco Challco[1],
Sheyla Fernandes[2], Jorge Coelho[4], Julio Costa[2], Yang Pian[3], Alexandre Marinho[1],
Seiji Isotani[5]

1 Center of Excellence for Social Technologies/Computing Institute, Federal University of Alagoas, Maceió, Alagoas, Brazil, 2 Institute of Psychology, Federal University of Alagoas, Maceió, Alagoas, Brazil, 3 Advanced Innovation Center for Future Education, Faculty of Education, Beijing Normal University, Beijing, China, 4 Faculty of Medicine, Federal University of Alagoas, Maceió, Alagoas, Brazil, 5 Institute of Mathematics and Computer Science, University of São Paulo, São Carlos, Brazil

* ig.ibert@ic.ufal.br (IIB); luyu@bnu.edu.cn (YL)

**Data Availability Statement:** All relevant data are within the manuscript and its Supporting information files.

## Abstract

"Flow experience" is a term used to describe the state of being fully immersed in what you are doing. The Flow State Scale (FSS-2) was developed to assess how people feel when they are in the flow state while participating in certain sports activities. The goal of this study was to obtain a short adapted version of the FSS-2 for the Brazilian-Portuguese language and for general activities (FSS-BR-S). To do this, we translated it both ways (forwards and backwards) and verified that the translation was accurate. Methods: After getting answers from 396 Brazilian participants, we performed (1) the construct validity of the FSS-BR-S and (2) the psychometric item quality analysis. The confirmatory factorial analysis shows that a FSS-BR-S factorial model is the best fit for the data ($\chi^2 = 44.36$, $p = .023$, $df = 27$, $\chi^2/df = 1.64$, CFI = 0.99, TLI = 0.98, and RMSEA = 0.04). Reliability tests done in this structure show that the FSS-BR-S (which only has nine items) has good internal consistency. The item quality analysis reveals that its difficulty and differentiating parameters are good for estimating the overall flow state. The test information curve for the short version demonstrates that it is very useful for estimating the flow states of each disposition. Discussion and Conclusions: Based on these findings, we can conclude that the FSS-BR-S has demonstrated sufficient validity to be used with Brazilians.

## Introduction

Positive psychology refers to the high levels of performance, enjoyment, and total immersion in a task as the flow experience [1, 2]. This experience was broadly documented in the literature [3], according to Kyriazos et al. [4], when he saw people pursuing particular occupations in a true state of total immersion. Csikszentmihalyi and Larson investigate this phenomena systematically in the context of intrinsically driven activities [5, 6]. The flow experience was subsequently characterized as a condition of complete absorption in an activity in which nothing else matters but the action itself, which is a pleasurable experience [1, 7].

**Funding:** The study was funded by Beijing Advanced Innovation Center for Future Education (AICFE) and the Conselho Nacional de Desenvolvimento Científico e Tecnológico (CNPq). Researchers from BNU participated in the data analysis, preparation of the manuscript and paper review.

**Competing interests:** The authors have declared that no competing interests exist.

In order to characterize the state of flow, later research identified this experience as a multi-dimensional process comprised of a concomitant occurrence of internal and external variables. These internal conditions include the individual's sense of interest in the work, their sense of well-being, and their detachment from reality. The external conditions are characterized by the existence of distinct objectives, unambiguous feedback, and a balance between perceived challenge and skill [7]. The nine underlying conditions were as follows: (1) challenge-skill balance; (2) merging of action and awareness; (3) clear goals; (4) unambiguous feedback; (5) concentration on the activity (task at hand); (6) sense of control; (7) loss of self-consciousness; (8) transformation of time by the loss of sense of time; and (9) autotelic experience. As depicted in Fig 1, a simple graphical representation of the flow state is a channel in which the challenge versus competence divides the states of anxiety and boredom. When the levels of challenge and skill are in equilibrium, the individual experiences an optimal experience or ideal state, depicted as a flow channel in the simplified model, resulting in a rise in performance [8].

One of the first methods developed to assess the flow's experience was the Experience Sampling Method (ESM) [9]. The Experience Sampling Method gathered evidence of nine factors, as well as the level of well-being experienced by participants, based on reports about experiences lived in natural environments. The protocol that guided this method took into account the weekly assessment of psychological states in which participants were stimulated to report about specific situations and their subjective status using a signaling device at randomly scheduled times. Initially, the instrument asked various questions about the level of response. Csikszentmihalyi then theoretically grounded a model that could be used to measure flow state in a scale presented in two versions of a psychometric instrument known as the long and short versions of the Flow State Scale (FSS) [7].

In the 1990s, Jackson and Marsh (1996) created a preliminary version of a structured questionnaire to assess the flow state experienced during specific sports activities [10]. Based on a nine-dimensional model, this instrument quantifies the flow state [7]. Jackson and Eklund

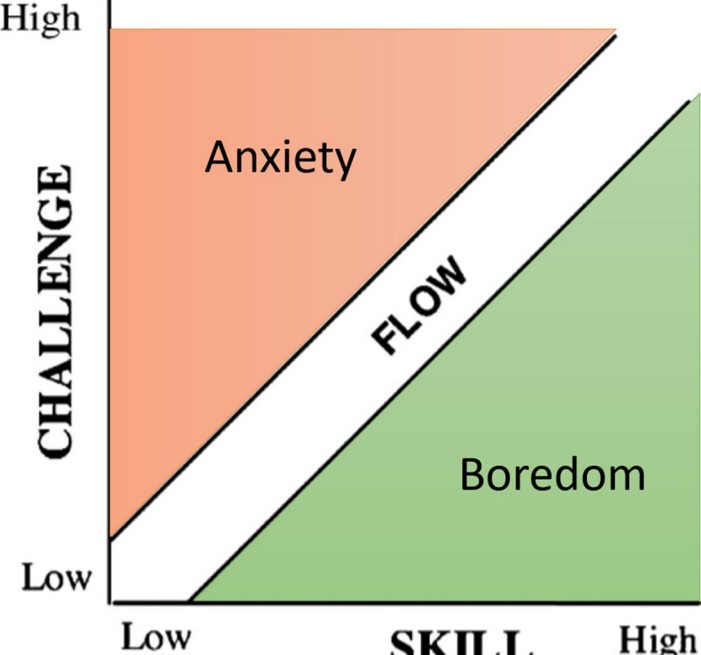

**Fig 1. Csikszentmihalyi's simplified model of flow experience.**

continued to adapt and develop the initial version, introducing the FSS-2 in the 2000s, which is now considered the measurement of flow status during an event [11]. The FSS-2 is used in the post-event to estimate the flow state of a recently completed activity. This instrument contained 36 items, and 13 items were added during validation to confirm construct validity [2, 10, 11].

Jackson et al. provided evidence to support the FSS-2 version as a nine-factor multi-correlated model [12]. The authors also tested a short version of the FSS-2 with only nine items, each of which corresponded to a dimension. This short version produced acceptable fit indexes, making it an appropriate alternative for measuring flow state in situations when the long version is too time consuming. As a result, the FSS-2 manual (in long and short version) was published by the authors [2]. Since then, the FSS-2 has been validated in a variety of contexts, in different languages, and for a variety of activities (e.g., as for learning [13], musical performance [14], sport activities [15], and people with schizophrenia or schizoaffective disorder [16]), presenting itself as a good consistent to perform cross-cultural studies in different nations [14–18].

Despite the fact that the FSS-2's authors provide a translated version of the FSS-2 for the Brazil-Portuguese language (from the website https://mindgarden.com/). There have been a few published studies in Brazil that demonstrate the validity of the translated FSS-2 and a shorter version of the FSS-2. It is necessary to ensure the conceptual and idiomatic equivalence of all of its items [19, 20]. The equivalence of what is measured and operationalized in the original language must be maintained in translated items [21]. A psychometric instrument's adaptation for a new culture should also take into account the relevance of concepts as well as the domains in which the instrument is used. In this sense, studies assessing validity should be conducted in the new culture, in different domains, with different populations, and using different versions of the same instruments to ensure the appropriateness of the items in terms of the ability of respondents to understand them (this latter for cross-cultural validation).

To the best of our knowledge, there is only one published study in English that validates an adaptation of the FSS-2 to the Brazil-Portuguese language for sport activities [15]. There is no published study on the validation of the shorter version of the FSS-2 for Brazilian. As a result, we conducted a study in which we presented the results of classical validation and psychometric item quality analysis for an adapted, shorter version of the FSS-2 for Brazilian contexts (FSS-BR-S) with nine items (FSS-BR-S).

## Methods

### Participants

Participants' responses were gathered after they performed different types of activities. These activities were indicated by the respondents in an open-ended question, and they were: solving logic problems in a gamified environment to measure aggressiveness (38.13%—151 responses), learning about physics (35.10%—139 responses), security information activities (12.37%—49 responses), using Kahoot! (3.03%—12 responses), solving quiz using Google Form (3.03%—12 responses), and other nine activities (8.29%, with less than 3% per activity).

In this study, we used a non-probabilistic sample (through volunteer sampling) of $n = 396$ Brazilians. Respondents had to be Brazilian citizens or foreigners living in Brazil who spoke Portuguese fluently. Inconsistent responses were eliminated from the data, and the respondents who provided them were not included in the sampling.

There were 205 males (51.77%), 183 females (46.21%), and eight participants (2.02%) who decided not to disclose their gender. Regarding participant age, the majority (179 participants, 45.20%) were between 15 and 18 years old, 85 participants (21.46%) were between 18 and 24

years old, 78 participants (19.70%) were between 25 and 34 years old, 30 participants (7.58%) were between 35 and 44 years old, ten participants (2.53%) were between 45 and 54 years old, seven participants (1.77%) were more than 54 years old, and seven participants (1.77%) decided not to participate. These participants' civil statuses were as follows: single (182, 92.86%), married (11, 5.61%), two (1.02%) stated to be in a stable relationship, and one (0.26%) was a widow. In terms of ethnicity, 128 individuals (65.31%) declared to be white, 50 (25.51%) stated to be pardo (mixed-race), five (2.55%) declared to be black, five (2.55%) declared to be Asian, and eight (4.08%) decided not to disclose their ethnicity. In terms of sexual orientation, 159 (81.12%) of the participants identified as heterosexual/straight, 12 (6.12%) as bisexual, four (2.04%) as gay, and 21 (10.71%) did not desire to identify their sexual orientation. 107 participants (54.59%) declared themselves to be from the middle economic class, 62 participants (31.63%) from the middle lower economic class, 15 participants (7.65%) from the lower economic class, ten participants (5.10%) from the middle higher economic class, and two participants (0.52%) from the higher economic class.

Participants' responses were collected after they performed various types of activities. Respondents to an open-ended question indicated the following activities: solving logic problems in a gamified environment (38.13%—151 responses), learning about physics (35.10%—139 responses), security information activities (12.37%—49 responses), using Kahoot! (3.03%—12 responses), solving quiz using Google Form (3.03%—12 responses), and other nine activities (8.29%).

### Recruitment

The data gathering procedure was completely carried out via the Internet, with respondents' voluntary participation. To acquire responses to the questionnaires used in this study, researchers distributed recruiting messages via their own social media networks (e.g., Facebook, Instagram, Whatsapp) and e-mails. There was no specific group targeted on social media networks. The invitation to participate in the survey did not employ lists that allow third parties to identify the visitors or visualize their contact data (e-mail, phone, etc.). After completing various activities, the participants completed the Brazilian Portuguese-translated version of the items corresponding to the FSS-2 questionnaire, which was made available in electronic form. When the questionnaire was halted, respondents might restart it later as long as they did not remove their browsing history. The form was open for 180 days, during which time we collected the data used in this study. During this time, the researchers involved were also available to answer any questions the participants had about the study.

### Ethics approval

We strictly adhered to the Brazilian National Health Council resolutions 466/12 and 510/16. (CNS). Furthermore, we strictly adhered to all of the standards outlined in Circular Letter No. 2/2021/CONEP/SECNS/MS. This research investigation was approved by the Human Research Ethics Committee of the Federal University of Alagoas (UFAL) (Protocol No. 35701820.3.0000.5013). As a result, the participants were informed that they were not required to participate in the research and that they might withdraw at any time if they were uncomfortable for any reason. Prior to answering the questionnaire, the participants (and their parents in the case of minors) agreed to a Free Prior and Informed Consent (FPIC) in which we informed the participants that their provided information would be confidential, with no way of individual identification, and that their responses would be analyzed as a whole rather than individually. Each participant spent around twenty (20) minutes reading the FPIC and answering the questionnaire.

## Procedure

We translated and adapted the original English version of Flow State Scale (FSS-2) [2]. The translation and adaption were completed in accordance with the International Test Commission's procedures [22]. Two separate interpreters performed the forward translation of items from the original English version prepared by Martin and Eklund (2002), amended by Jackson, Martin, and Eklund (2008) [12], and published in 2010 [2]. A third independent translator then reversed the translation from Brazilian Portuguese to English. This semantic assessment was carried out by ten persons to determine whether the instrument could be comprehended by the people for whom it was designed. This assessment was completed in two stages, with five participants participating in each stage. Participants were initially asked to read the instrument item by item. After reading each item, participants were asked about their understanding of the item in the second stage. When there was a doubt regarding words or sentences, it was recorded and corrections were made, but no significant changes were required.

## Instruments

The instrument used to collect the data for this study was the 36-item translated to Brazilian-Portuguese version of the FSS-2 (FSS-BR). As part of the dataset, participants' age, gender, ethnicity, sexual orientation, marital status, and socioeconomic status were collected using a demographic questionnaire. Items on the FSS-BR utilized a 5-point Likert scale ranging from 1 (strongly disagree) to 5 (strongly agree), with four items for each of the nine components, also known as factors or dimensions: Challenge-Skill Balance (CSB), Merging of Action-Awareness (MAA), Clear Goals (CG), Unambiguous Feedback (UF), Concentration on Task at Hand (CTH), Sense of Control (SC), Loss of Self-Consciousness (LSC), Transformation of Time (TT), and Autotelic Experience (AE).

## Data analysis procedure

The dataset was prepared using Google Spreadsheets, and then it was analyzed using R Software [23] to seek evidence of the construct validity of the FSS-BR and its short versions (FSS-BR-S). To determine its construct validity, the structure, internal consistency, convergence, and discrimination were assessed. After demonstrating this validation, we utilized the data to conduct a quality analysis (difficulty and discriminating) of questions based on Item Response Theory (IRT).

We cannot conduct Multi-Group Confirmatory Factor Analysis (MGCF) to determine whether the FSS-2 has the same or different response patterns across different Brazilian populations because we did not collect information on the respondents' residences and we did not conduct a probabilistic sample to obtain respondents from the five regions of Brazil. Due to the same constraints, we are unable to conduct a Differential Item Functioning (DIF) analysis to compare the endorsement likelihood of the different Brazilian populations.

Lavaan package [24] was used to do the Confirmatory Factor Analysis (CFA) to determine the structure and item loadings for the construct validity. CFA was performed using the Weighted Least Squares Mean and Variance-Adjusted (WLSMV) estimator with polychoric matrices due to the ordinal character of the dataset and its non-normal distribution [25]. This estimator has been shown to be effective for ordinal data, and it does not necessitate a normal distribution to estimate the fit indices [26, 27].

Two models were used to evaluate the structures of the short version of the FSS-BR-S in the CFA: the unidimensional model with the original 9 items published in the manual of Jackson et al. (2010) [2]; and an alternative unidimensional model that includes items with greater discrimination power—the ability to differentiate between individuals with a low overall flow

state and those with a high overall flow state. In other words, to select items for the alternative unidimensional model of the FSS-BR-S, the discrimination parameter (a) from an Item Response Theory (IRT) analysis was used to provide an index of how well the item distinguishes between high and low flow states. Since the objective of most psychometric measures is to differentiate between examinees' responses, items with a higher discrimination parameter (a) are typically considered to be superior. Items with low parameters provide minimal measurement data and may be evaluated for replacement or elimination [28]. The method we utilized will allow future studies to do criterion validity tests, which are the gold standard for identifying the degree of goodness with which an instrument's results represent its measurement [29].

Lavaan Package [24] was utilized to perform the CFA without requiring the setting of preset parameters, starting values, modifiers, or error values. The following parameters of the adjustment indices were used to examine the outcomes in order to evaluate the model's quality:

- Chi-square/degrees of freedom ($\chi^2/df$) in which values less than 2 indicated an excellent fit, values between 2 and 3 suggested a good fit, and values up to 5 indicated an acceptable fit [30, 31].

- The Comparative Fit Indicator (CFI) is an additional model adjustment index where values equal to or better than 0.95 indicate a valid model [32].

- The Tucker Lewis Index (TLI) is a model modification comparison index where values of 0.95 or higher are judged appropriate [32].

- The Root Mean Square Error of Approximation (RMSEA), which is defined as the difference between the model and the observed data, has values less than or equal to 0.05, indicating a great fit [32].

Although Niemand and Mai (2018) [33] claimed that researchers typically err when evaluating structural equation models using "golden rules of thumb," this is not always the case. Universal cutoffs are problematic because they ignore sample size, factor loadings, the number of latent variables and items, and whether or not the provided data have a normal distribution [33]. Thus, in this study, the aforementioned values are just considered as indicative of an ideal model.

The internal consistency was evaluated using the semTool software [34], and in addition to the standard test indications such as Cronbach's alpha (*alpha*), we calculated McDonald's omega (*omega*) [35] and Composite Reliability (CR) [36]. McDonald's *omega* and CR employ congeneric models that overcome the underestimate produced in Cronbach's *alpha* by the assumption of tau-equivalence in all items. The threshold levels for these internal consistency measurements were defined between values of 0.70 and 0.95 [29].

In terms of convergent and discriminant validity, we looked at the values of CR and the Average Variance Extracted (AVE) [37], as well as the heterotrait-monotrait (HTMT) ratio [38]. These indices were produced with the semTools package [34], and for convergent validity, we used the following criteria:

- AVE values more than 0.5 indicated good convergence factor [39]; and

- convergence was still satisfactory if AVE was less than 0.50 but the CR of all factors were near to or greater than 0.60 [37].

We looked at the AVE values and factor correlations to determine discriminant validity. According to the literature [40], there is a significant discriminant of factors if the square root of the AVEs is bigger than the correlation coefficients between the factors. This approach

indicates discriminant problems when there are no true issues [41]. As a result, we supplemented the evaluation of AVE and the correlation variables with the $HTMT_{0.85}$ criterion, in which values close to or less than 0.85 indicate acceptable validity [42].

Individual psychometric item quality analyses were performed using IRT analysis and the Mirt software [43] (specifically, the graded package). The IRT was designed as an alternative to the Classical Test Theory (CTT), and it is an item quality framework in which difficulty and discriminating item factors explain the relationships of item answers within and between people [44]. In comparison to the CTT, the IRT overcomes the difficulties of assessing individual item parameters as well as the unrealistic assumption that confidence intervals for a person's latent construct are uniform [45]. In sense, we estimate the difficulty and discrimination parameters for all FSS-2 items using IRT and Graded Response Models (GRM) [46]. This model is a mathematical model family that in IRT analysis deals with ordered polytomous categories such as strongly disagree, disagree, agree, and highly agree, which are utilized in attitude surveys. The Test Information Curve (TIC) was also used in the item quality study to determine the latent trace intervals where the instrument is more accurate in producing any level of participant ability (in our case the level of flow state) [47].

## Results

Table 1 provides descriptive statistics (M: Mean, Mdn: Median, and SD: Standard Deviation) for all data collected. The table also includes the Measure of Sampling Adequacy (MSAi) for all the items, as well as their normality validity, which includes Skewness (Skew), Kurtosis (Kurt), the Statistic of Shapiro-Wilk test and p-value.

### Construct validity of the long version of the FSS-2

**Structure validity.** Table 2 presents the CFA fit indices on the models used to test the structure validity of the FSS-2's long version. Although this is not the primary goal in our research, it is a required step in order to select the nine items for the short version proposed here (FSS-BR-S). According to the $\chi^2/df$, the models exhibit an acceptable fit (between 3 and 5), with the multi-correlated model (with $\chi^2 = 2369.826$, and $p =< 0.001$) slightly outperforming the second-order model (with $\chi^2 = 2771.145$, and $p =< 0.001$). The CFI and TLI were both less than the optimal cutoff level (0.95), and the RMSEAs were greater than the ideal cutoff limit (0.05).

Fig 2 depicts the item loadings, covariances, and residuals for the correlated model of nine factors: Challenge Skill Balance (CSB), Autotelic Experience (AE), Transformation of Time (TT), Loss of Self-Consciousness (LSC), Sense of Control (SC), Concentration on the Task at Hand (CTH), Unambiguous Feedback (UF), Clear Goals (CG), and Merging of Action and Awareness (CA) (MAA). It is important to note that the majority of item loadings have values more than 0.30, with the exceptions of items Q6 ($\lambda = 0.22$) and Q12 ($\lambda = 0.23$). S1 Appendix details the questions used as items in our translated version of the FSS-2.

Table 3 shows the covariances on the multi-correlated model of nine-factors. The majority of pairs exhibit moderate (0.50 to 0.70) and high (0.70 to 0.90) correlations. Unambiguous Feedback (UF) vs. Merging of Action-Awareness (MAA) and Unambiguous Feedback (UF) vs. Transformation of Time (TT) had lower correlation values with values of 0.251 and 0.232, respectively. The correlations of factors are consistent with the theory upon which the FSS-2 questionnaire was based, a multi-correlated model comprising nine factors.

**Internal consistency.** Table 4 displays the results of internal consistency tests done on the long version of FSS-2. In general, the Cronbach's $\alpha$, McDonald's $\omega$, and CR values for the composition of all 36 items were more than 0.80, showing excellent internal consistency to measure

**Table 1. Descriptive statistics, normality test, and MSA of the dataset.**

| Items | M | Mdn | SD | Skew | Kurt | Statistic | p-value | MSAi |
|---|---|---|---|---|---|---|---|---|
| Q1 | 3.987 | 4 | 0.831 | -0.960 | 1.368 | 0.814 | <.001 | 0.901 |
| Q2 | 3.167 | 3 | 1.164 | 0.123 | -0.919 | 0.906 | <.001 | 0.891 |
| Q3 | 3.306 | 3 | 0.981 | -7.798 | -0.738 | 0.895 | <.001 | 0.884 |
| Q4 | 3.717 | 4 | 0.886 | 0.000 | 0.164 | 0.862 | <.001 | 0.909 |
| Q5 | 3.551 | 4 | 0.989 | -0.202 | -0.452 | 0.886 | <.001 | 0.902 |
| Q6 | 3.136 | 3 | 1.225 | 0.123 | -1.062 | 0.895 | <.001 | 0.841 |
| Q7 | 3.515 | 4 | 1.032 | -1.644 | -0.433 | 0.885 | <.001 | 0.902 |
| Q8 | 3.071 | 3 | 1.199 | 0.100 | -0.873 | 0.914 | <.001 | 0.688 |
| Q9 | 3.505 | 4 | 1.032 | -0.142 | -0.238 | 0.890 | <.001 | 0.800 |
| Q10 | 3.793 | 4 | 1.003 | 0.123 | -0.217 | 0.873 | <.001 | 0.892 |
| Q11 | 3.414 | 4 | 1.041 | -1.153 | -0.596 | 0.897 | <.001 | 0.919 |
| Q12 | 3.076 | 3 | 1.048 | 0.249 | -0.790 | 0.906 | <.001 | 0.838 |
| Q13 | 3.167 | 3 | 0.992 | -0.598 | -0.756 | 0.898 | <.001 | 0.849 |
| Q14 | 3.598 | 4 | 1.120 | 0.123 | -0.309 | 0.867 | <.001 | 0.913 |
| Q15 | 3.639 | 4 | 0.910 | -4.861 | -0.250 | 0.873 | <.001 | 0.909 |
| Q16 | 3.331 | 4 | 1.156 | 0.000 | -0.989 | 0.896 | <.001 | 0.799 |
| Q17 | 3.278 | 3 | 1.018 | -0.425 | -0.632 | 0.903 | <.001 | 0.846 |
| Q18 | 3.043 | 3 | 1.082 | 0.123 | -0.710 | 0.915 | <.001 | 0.810 |
| Q19 | 3.553 | 4 | 1.024 | -3.457 | -0.447 | 0.892 | <.001 | 0.851 |
| Q20 | 3.530 | 4 | 1.124 | 0.001 | -0.730 | 0.895 | <.001 | 0.815 |
| Q21 | 3.366 | 3 | 0.978 | -0.229 | -0.364 | 0.900 | <.001 | 0.902 |
| Q22 | 3.242 | 3 | 1.017 | 0.123 | -0.867 | 0.879 | <.001 | 0.853 |
| Q23 | 3.510 | 4 | 0.997 | -1.861 | -0.433 | 0.891 | <.001 | 0.884 |
| Q24 | 3.682 | 4 | 0.900 | 0.063 | -0.395 | 0.873 | <.001 | 0.923 |
| Q25 | 3.386 | 4 | 1.031 | -0.492 | -0.701 | 0.888 | <.001 | 0.893 |
| Q26 | 3.323 | 3 | 1.070 | 0.123 | -0.684 | 0.900 | <.001 | 0.882 |
| Q27 | 3.442 | 4 | 0.978 | -3.996 | -0.367 | 0.897 | <.001 | 0.916 |
| Q28 | 3.232 | 3 | 1.152 | 0.000 | -0.875 | 0.912 | <.001 | 0.827 |
| Q29 | 3.043 | 3 | 1.107 | -0.128 | -0.886 | 0.909 | <.001 | 0.648 |
| Q30 | 3.841 | 4 | 0.959 | 0.123 | -0.399 | 0.866 | <.001 | 0.927 |
| Q31 | 3.578 | 4 | 0.926 | -1.038 | -0.410 | 0.875 | <.001 | 0.876 |
| Q32 | 3.669 | 4 | 1.053 | 0.299 | -0.492 | 0.869 | <.001 | 0.863 |
| Q33 | 3.702 | 4 | 0.867 | -0.520 | 0.054 | 0.850 | <.001 | 0.936 |
| Q34 | 3.409 | 4 | 1.234 | 0.123 | -0.930 | 0.892 | <.001 | 0.796 |
| Q35 | 3.265 | 3 | 1.142 | -4.225 | -0.760 | 0.910 | <.001 | 0.770 |
| Q36 | 3.846 | 4 | 1.053 | 0.000 | -0.386 | 0.862 | <.001 | 0.868 |

M: mean; Mdn: Median; SD: Standard Deviation; Skew: Sweetness; Kurt: Kurtosis; MSAi: Measure of Sampling Adequacy Index

**Table 2. Fit indexes of tested models in the validation of FSS-BR.**

| Model | $\chi^2$ | df | $\chi^2$/df | p-value | CFI | TLI | RMSEA [CI 95%] |
|---|---|---|---|---|---|---|---|
| Multi-corr | 2369.826 | 558 | 4.247 | <.001 | 0.871 | 0.854 | 0.091 [0.087, 0.094] |
| 2nd-order | 2771.145 | 585 | 4.737 | <.001 | 0.844 | 0.832 | 0.097 [0.094, 0.101] |

$\chi^2$: chi-square; df: degrees of freedom; CFI: Comparative Fit Index; TLI: Tucker-Lewis Index; RMSEA: Root Mean Square Error of Approximation.

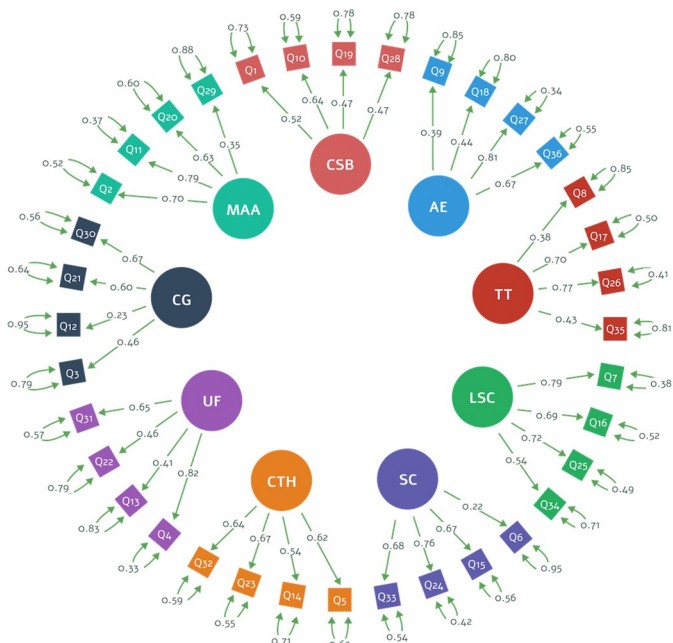

**Fig 2. Structure of the FSS-2 (multi-correlated model of 9-factors).**

the flow state. The factors, on the other hand, had internal consistency indices ranging from 0.55 to 0.77. For the factors Merging of Action and Awareness (MAA), Unambiguous Feedback (UF), Concentration on the Task at Hand (CTH), and Loss of Self-Consciousness (LSC), the majority of factors show internal consistency values near to 0.70s. Other components—Challenge Skill Balance (CSB), Clear Goals (CG), Sense of Control (SC), Transformation of Time (TT), and Autotelic Experience (AE) had internal consistency indices that were less than the suggested minimum (0.70).

Based on the findings of the internal consistency tests, we observe that the FSS-2 has internal consistency issues when measuring some of nine factors, including the Challenge Skill Balance (CSB), Clear Goals (CG), Sense of Control (SC), Time Transformation (TT), and Autotelic Experience (AE).

**Table 3. Covariances in the multi-correlated model of 9-factors.**

|       | CSB   | MAA   | CG    | UF    | CTD   | SC    | LSC   | TT    | AE  |
|-------|-------|-------|-------|-------|-------|-------|-------|-------|-----|
| CSB   | -     |       |       |       |       |       |       |       |     |
| MAA   | 0.513 | -     |       |       |       |       |       |       |     |
| CG    | 0.898 | 0.470 | -     |       |       |       |       |       |     |
| UF    | 0.672 | 0.251 | 0.937 | -     |       |       |       |       |     |
| CTH   | 0.725 | 0.561 | 0.708 | 0.579 | -     |       |       |       |     |
| SC    | 0.797 | 0.430 | 0.946 | 0.834 | 0.936 | -     |       |       |     |
| LSC   | 0.608 | 0.418 | 0.576 | 0.452 | 0.677 | 0.578 | -     |       |     |
| TT    | 0.617 | 0.654 | 0.469 | 0.232 | 0.646 | 0.516 | 0.459 | -     |     |
| AE    | 0.890 | 0.647 | 0.593 | 0.383 | 0.668 | 0.591 | 0.541 | 0.860 | -   |

CSB: Challenge Skill Balance; MAA: Merging of Action and Awareness; CG: Clear Goals; UF: Unambiguous Feedback; CTH: Concentration on the Task at Hands; SC: Sense of Control; LSC: Loss of Self-Consciousness; TT: Transformation of Time; and AE: Autotelic Experience.

**Table 4. Results of the internal reliability tests for the FSS-BR.**

|   | CSB | MAA | CG | UF | CTH | SC | LSC | TT | AE | total |
|---|-----|-----|-----|-----|-----|-----|-----|-----|-----|-----|
| $\alpha$ | 0.594 | 0.723 | 0.604 | 0.712 | 0.698 | 0.614 | 0.777 | 0.659 | 0.669 | 0.906 |
| $\omega$ | 0.596 | 0.717 | 0.556 | 0.673 | 0.709 | 0.636 | 0.773 | 0.655 | 0.671 | 0.921 |
| CR | 0.603 | 0.720 | 0.565 | 0.687 | 0.712 | 0.685 | 0.783 | 0.670 | 0.678 | 0.857 |

$\alpha$: Cronbach's alpha; $\omega$: McDonald's Omega; CR: Composite Reliability; CSB: Challenge Skill Balance; MAA: Merging of Action and Awareness; CG: Clear Goals; UF: Unambiguous Feedback; CTH: Concentration on the Task at Hands; SC: Sense of Control; LSC: Loss of Self-Consciousness; TT: Transformation of Time; and AE: Autotelic Experience.

**Convergent validity and discriminant validity.** The indicators used to test the convergent and discriminant validity of the FSS-2 are shown in Tables 5 and 6.

Table 5 shows the Composite Reliability (CR), Average Variance Extracted (AVE), and Variance Inflation Factor (VIF) values. For all factors, the AVE values were less than 0.50. In this sense, the CR values for all factors should be at least 0.60 to achieve an adequate convergence. Four of the nine factors had CR values larger than 0.70: Merging of Action and Awareness (MAA), Concentration on the Task at Hand (CTH), and Loss of Self-Consciousness (LSC). The other four of nine factors had CR values of 0.60s, and only the Clear-Goals (CG) had a CR value less than 0.60, suggesting that the FSS-2 has an adequate convergent validity.

Table 6 shows the relationship between the square root of the Average Variance Extracted (AVE) values (diagonal portion of the matrix), the factor correlations (bottom triangular part of the matrix), and the heterotrait-monotrait (HTMT) ratios (upper triangular part of the matrix). According to the Fornell-Larcker criterion [37], the square root of Average Variance Extracted (AVE) values should be larger than the factor correlations to indicate excellent discriminant validity. The results show that discrimination validity is good, and probable discriminant issues are highlighted as cursive numbers in the lower triangular area of Table 6. The discriminant validity was evaluated by looking at the heterotrait-monotrait (HTMT) ratios. Using the heterotrait-monotrait (HTMT) criteria given by Ab Hamid et al. (2017) [42], we can see that there are few potential discriminant difficulties indicated as cursive numbers. The potential issue can be seen in the pairs Challenge-Skill Balance (CSB) vs. Autotelic Experience (AE), Clear Goals (CG) vs. Unambiguous Feedback (UF), Clear Goals (CG) vs. Sense of Control (SC), and Concentration on Task at Hand (CTH) vs. Sense of Control (SC) with ratio values greater than 0.90.

## IRT analysis of the long version of the FSS-2

The item quality analysis based on IRT was carried out in the long version of the FSS-2 using the Graded Response Model (GRM).

**Table 5. Indexes of CR, AVE and VIF for the FSS-BR.**

|   | CSB | MAA | CG | UF | CTH | SC | LSC | TT | AE |
|---|-----|-----|-----|-----|-----|-----|-----|-----|-----|
| CR | 0.603 | 0.720 | 0.565 | 0.687 | 0.712 | 0.685 | 0.783 | 0.670 | 0.678 |
| AVE | 0.274 | 0.404 | 0.257 | 0.350 | 0.379 | 0.322 | 0.461 | 0.335 | 0.353 |
| VIF | 2.111 | 1.416 | 2.244 | 2.057 | 2.101 | 2.507 | 1.489 | 1.763 | 2.142 |

CR: Composite Reliability; AVE: Average Variance Extracted; VIF: Variance Inflation Factor; CSB: Challenge Skill Balance; MAA: Merging of Action and Awareness; CG: Clear Goals; UF: Unambiguous Feedback; CTH: Concentration on the Task at Hands; SC: Sense of Control; LSC: Loss of Self-Consciousness; TT: Transformation of Time; and AE: Autotelic Experience.

**Table 6. Factor correlations and heterotrait-monotrait (HTMT) ratios for the FSS-2.**

|  | CSB | MAA | CG | UF | CTH | SC | LSC | TT | AE |
|---|---|---|---|---|---|---|---|---|---|
| CSB | **0.523** | 0.547 | 0.853 | 0.682 | 0.727 | 0.841 | 0.630 | 0.661 | *0.960* |
| MAA | 0.513 | **0.635** | 0.498 | 0.327 | 0.570 | 0.469 | 0.411 | 0.674 | 0.617 |
| CG | 0.898 | 0.470 | **0.507** | *1.015* | 0.676 | *1.001* | 0.522 | 0.438 | 0.594 |
| UF | 0.672 | 0.251 | 0.937 | **0.592** | 0.554 | 0.887 | 0.416 | 0.257 | 0.441 |
| CTH | 0.725 | 0.561 | 0.708 | 0.579 | **0.616** | *0.982* | 0.688 | 0.621 | 0.641 |
| SC | 0.797 | 0.430 | 0.946 | 0.834 | 0.936 | **0.567** | 0.581 | 0.534 | 0.659 |
| LSC | 0.608 | 0.418 | 0.576 | 0.452 | 0.677 | 0.578 | **0.679** | 0.459 | 0.542 |
| TT | 0.617 | 0.654 | 0.469 | 0.232 | 0.646 | 0.516 | 0.459 | **0.578** | 0.898 |
| AE | 0.890 | 0.647 | 0.593 | 0.383 | 0.668 | 0.591 | 0.541 | 0.860 | **0.594** |

CSB: Challenge Skill Balance; MAA: Merging of Action and Awareness; CG: Clear Goals; UF: Unambiguous Feedback; CTH: Concentration on the Task at Hands; SC: Sense of Control; LSC: Loss of Self-Consciousness; TT: Transformation of Time; and AE: Autotelic Experience. The bold numbers listed diagonally are the square root of Average Variance Extracted (AVE) values. The correlation among the factors are located in the lower part of the table, and the heterotrait-monotrait (HTMT) ratio of the correlation coefficients are located in the upper part.

Table 7 shows the discrimination and difficulty parameters found for all 36 items in the FSS-2 questionnaire's long version. The item discrimination mean, parameter (**a**), is $M = 1.207$ with a standard deviation of 0.492, indicating a moderate level of discrimination (between 0.65 and 1.34). This discrimination value demonstrates that the FSS-2 has sufficient power to distinguish individuals with similar levels of flow state. Q24 has the highest discrimination value of 2.405 (very-high), and the items with the lowest discrimination values are Q12, Q6, Q29, Q8, and Q13, with values of 0.365, 0.449, 0.456, 0.535, and 0.542, respectively.

Regarding the difficulty parameters of items ($b1$—$b4$ and $bx$), as the location where the majority of people have their responses, we assessed the response thresholds. The results show that no single difficulty average ($bx$) falls outside of the [−3; +3] range, which is defined as adequate for the IRT metric. The average of the difficulty parameters ($bx$) is in the low levels of the latent factor (mean $M = -0.781$ and $SD = 0.342$). The item Q1 with $bx = -1.711$ has the lowest difficulty average value, indicating that this item is more easily endorsed by participants who have lower flow state (less difficult). The item Q18 has the highest difficulty average value of $bx = -0.112$, which is considered moderate difficulty and is far from the upper limit defined in the IRT metric (+ 3). Based on these values, we can conclude that the majority of endorsement (difficulty parameters) were located below the mean of flow state (right-skewed latent trait).

For the future versions of the FSS-2, one suggestion is that these items should be improved semantically to avoid this skewness in the distribution of responses.

Difficulty parameter values of items outside of the range considered adequate were indicated in Table 7 with italic text. Item levels with difficulty values less than −3 are items frequently preferred by participants, while item levels with difficulty values greater than + 3 indicate item levels with fewer responses. One suggestion for future versions of the FSS-2 is that these items should be modified semantically to avoid skewness in response distribution.

Fig 3 depicts the Test Information Curve for assessing the accuracy of the long version of the FSS-2 using IRT analysis. This curve graph reveals that the greatest quantity of information is measured between −4 and + 3, indicating that the FSS-2 is accurate (reliable) when measuring the flow state in individuals with a theta (*theta*) within this range.

**Table 7. Discrimination and difficulty item parameters for the long version of FSS-2.**

| | a | b1 | b2 | b3 | b4 | bx |
|---|---|---|---|---|---|---|
| Q1 | 1.296 | -3.883 | -2.631 | -1.334 | 1.006 | -1.711 |
| Q2 | 1.144 | -2.481 | -0.945 | 0.079 | 2.070 | -0.319 |
| Q3 | 0.871 | -4.775 | -1.522 | 0.140 | 2.827 | -0.833 |
| Q4 | 1.839 | -3.017 | -1.684 | -0.583 | 1.262 | -1.006 |
| Q5 | 1.429 | -3.235 | -1.440 | -0.308 | 1.568 | -0.854 |
| Q6 | 0.449 | -4.835 | -1.601 | 0.112 | 4.494 | -0.457 |
| Q7 | 1.421 | -2.916 | -1.370 | -0.331 | 1.560 | -0.764 |
| Q8 | 0.535 | -3.869 | -1.515 | 0.855 | 3.801 | -0.182 |
| Q9 | 0.742 | -4.466 | -2.392 | -0.426 | 2.485 | -1.200 |
| Q10 | 1.630 | -2.987 | -1.734 | -0.560 | 0.809 | -1.118 |
| Q11 | 1.408 | -2.827 | -1.232 | -0.172 | 1.666 | -0.641 |
| Q12 | 0.365 | -7.800 | -2.156 | 1.218 | 7.040 | -0.424 |
| Q13 | 0.542 | -6.455 | -1.843 | 0.640 | 4.863 | -0.699 |
| Q14 | 1.370 | -2.524 | -1.398 | -0.617 | 1.235 | -0.826 |
| Q15 | 1.725 | -3.289 | -1.569 | -0.439 | 1.374 | -0.981 |
| Q16 | 1.174 | -2.909 | -0.954 | -0.068 | 1.674 | -0.564 |
| Q17 | 1.353 | -2.874 | -1.133 | 0.103 | 2.049 | -0.464 |
| Q18 | 0.769 | -3.475 | -1.108 | 0.785 | 3.352 | -0.112 |
| Q19 | 0.945 | -4.147 | -1.984 | -0.412 | 1.840 | -1.176 |
| Q20 | 1.080 | -3.355 | -1.579 | -0.366 | 1.337 | -0.991 |
| Q21 | 1.334 | -3.008 | -1.399 | 0.059 | 1.969 | -0.595 |
| Q22 | 0.673 | -5.290 | -1.548 | 0.016 | 3.953 | -0.717 |
| Q23 | 1.668 | -2.775 | -1.341 | -0.263 | 1.436 | -0.736 |
| Q24 | 2.405 | -3.047 | -1.417 | -0.432 | 1.078 | -0.955 |
| Q25 | 1.319 | -3.116 | -1.181 | -0.166 | 1.911 | -0.638 |
| Q26 | 1.466 | -2.547 | -1.070 | -0.046 | 1.752 | -0.478 |
| Q27 | 2.021 | -2.426 | -1.238 | -0.101 | 1.434 | -0.583 |
| Q28 | 0.892 | -3.345 | -1.202 | 0.366 | 2.067 | -0.529 |
| Q29 | 0.456 | -5.829 | -1.428 | 1.049 | 5.106 | -0.275 |
| Q30 | 1.879 | -3.270 | -1.643 | -0.640 | 0.738 | -1.204 |
| Q31 | 1.074 | -4.706 | -1.941 | -0.453 | 2.022 | -1.269 |
| Q32 | 1.591 | -2.936 | -1.332 | -0.565 | 1.072 | -0.940 |
| Q33 | 1.815 | -3.381 | -1.637 | -0.609 | 1.399 | -1.057 |
| Q34 | 0.763 | -3.490 | -1.457 | -0.257 | 1.862 | -0.836 |
| Q35 | 0.636 | -4.230 | -1.782 | 0.233 | 2.971 | -0.702 |
| Q36 | 1.383 | -3.323 | -1.801 | -0.686 | 0.656 | -1.288 |
| M | 1.207 | -3.690 | -1.534 | -0.116 | 2.215 | -0.781 |
| SD | 0.492 | 1.171 | 0.367 | 0.526 | 1.392 | 0.342 |

M: mean; SD: Standard Deviation.

## Construct validity of the short version of the FSS-2

Employing items from the long version of the FSS-2, Jackson et al. (2010) [2] developed and validated a nine-item flow state measurement instrument based on a unidimensional model. These items translated from English to Brazilian-Portuguese constitute the FSS-2 original short version. Table 8 provides a summary of the structure's fit indices, which were calculated

## Test Information and Standard Errors

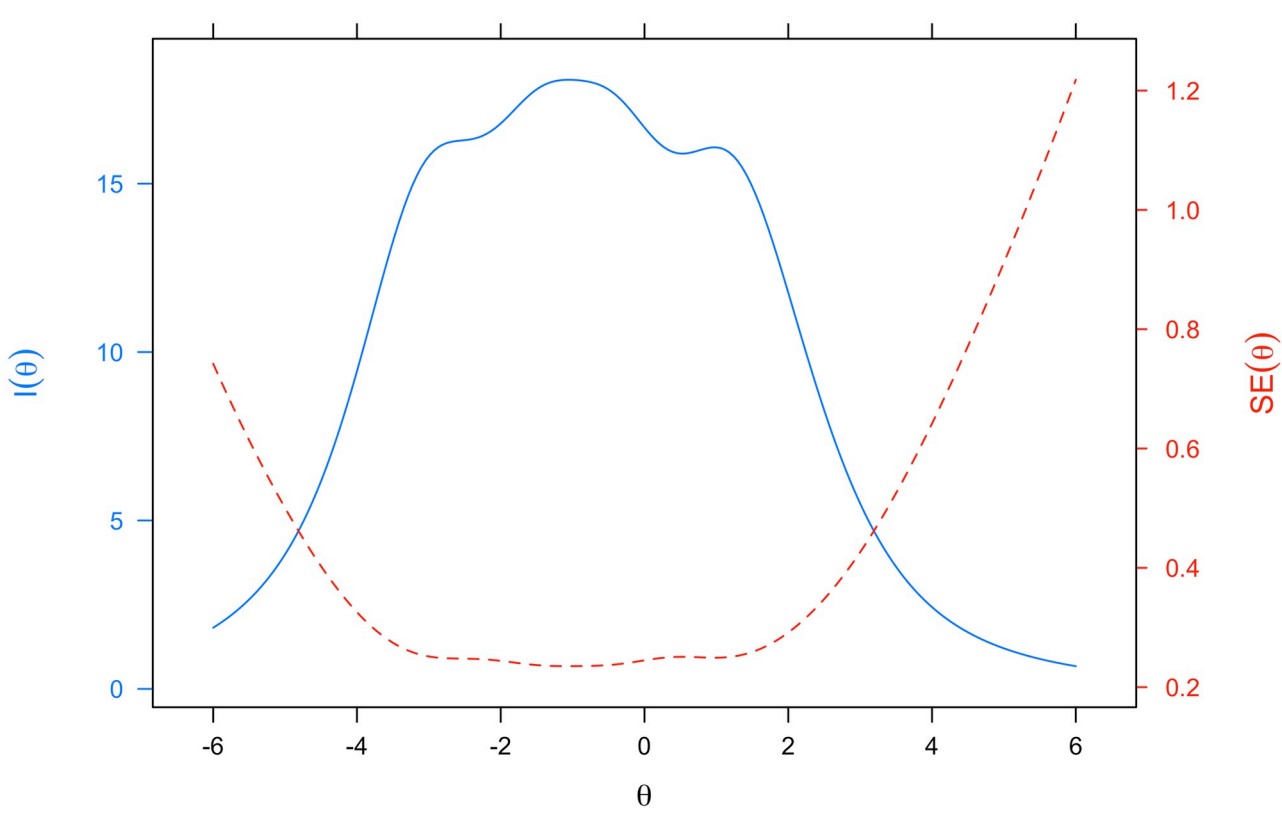

**Fig 3. Test information curve of the long version of FSS-2.**

using CFA and a unidimensional model to evaluate its structure validity. This model does not have appropriate fitting values because $\chi^2/df$ is greater than 5, CFI is less than 0.95, TLI is less than 0.95, and RMSEA is greater than 0.10. Due to these unsatisfactory fitting indices, we developed a shortened version of the FSS-2 as an alternative to the original version, known as FSS-BR-S. The parameter "**a**" from the IRT analysis in the long version of the FSS-2 was utilized to choose items for the FSS-BR-S alternative unidimensional model based on their ability to distinguish between low and high values of flow states. Most psychometric measurement instruments are designed to differentiate responses; hence, items with greater discriminating values (parameter "a") are preferable. Consequently, item Q10 replaced item Q19 for the factor Challenge-Skill Balance (CSB); item Q11 replaced item Q29 for the factor Merging of Action-Awareness (MAA); item Q30 replaced item Q12 for the factor Clear Goals (CG); item Q4 replaced item Q22 for the factor Unambiguous Feedback (UF); item Q23 replaced item Q32

**Table 8. Fit indexes of the CFA in the original and alternative short versions of the FSS-2.**

| Model | $\chi^2$ | df | $\chi^2/df$ | p-value | CFI | TLI | RMSEA [CI 95%] |
|---|---|---|---|---|---|---|---|
| original-s | 305.505 | 27 | 11.315 | <.001 | 0.539 | 0.385 | 0.162 [0.146, 0.178] |
| alternative-s | 44.361 | 27 | 1.643 | 0.023 | 0.990 | 0.986 | 0.040 [0.017, 0.061] |

$\chi^2$: chi-square; df: degrees of freedom; CFI: Comparative Fit Index; TLI: Tucker-Lewis Index; RMSEA: Root Mean Square Error of Approximation.

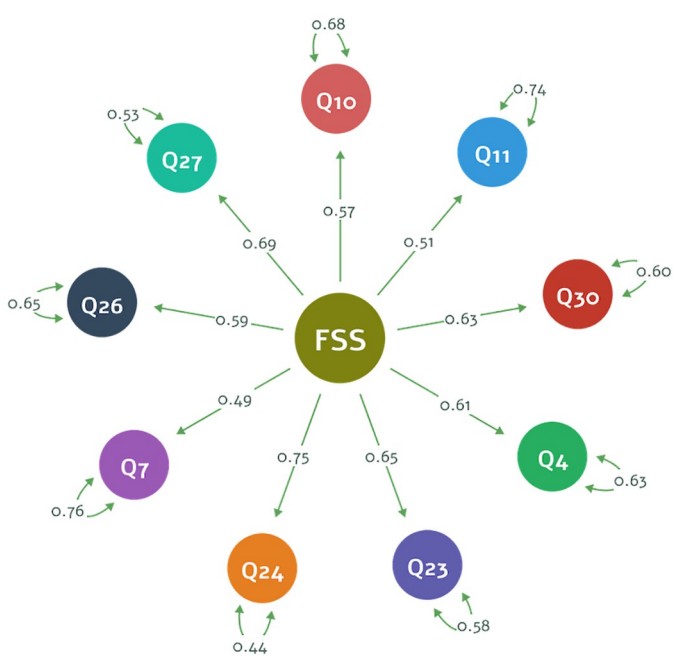

**Fig 4. Structure of the FSS-BR-S (unidimensional model of 9 items).**

for the factor Concentration on Task at Hand (CTH); item Q24 replaced item Q6 for the factor Sense of Control (SC); item Q7 was preserved for the factor Loss of Self-Consciousness (LSC); item Q26 replaced item Q17 for the factor Transformation of Time (TT); and item Q27 replaced item Q36 for the factor Autotelic Experience (AE).

The CFA demonstrates excellent fits in the $\chi^2/df$ index (less than 2) for the alternative short version of the FSS-2 as displayed in Table 8. CFI had value more than 0.95s, TLI was greater than 0.95s, and RMSEA was less than 0.05 indicating that our alternative model has good-fitting values. These indexes indicate that the empirical data are most compatible with the alternative version, which we will refer to as FSS-BR-S and its questions used as items are detailed in S2 Appendix. Fig 4 displays the FSS-BR-S item loadings and residuals. In this structure, there are no statistically significant differences in the item loadings, indicating that the FSS-BR-S extracts nearly identical variances from each of the nine items and that none of these items substantially changed the variance in the measurement of flow state.

Regarding the internal consistency of the FSS-BR-S, the reliability tests results were Cronbach's $\alpha$ = 0.839 and McDonald's $\omega$ = 0.840. These values indicate excellent internal consistency for the alternative short version of the FSS-2 using the unidimensional model.

## IRT analysis of the FSS-BR-S

Table 9 displays the estimated discrimination and difficulty parameters for the FSS-BR-S. These parameters were estimated using Graded Response Models, and the item discrimination mean, parameter (**a**), is $M$ = 1.686 with $SD$ = 0.416, showing a high level of discrimination (from 1.35 to 1.69). This evidence demonstrates that the FSS-BR-S has the ability to distinguish individuals with similar level values of flow state. The item with the highest discrimination value was Q24, with a value of 2.612 (very-high), while no item has a discrimination value below 0.70.

**Table 9. Discrimination and difficulty parameters for the FSS-Short.**

|     | a | b1 | b2 | b3 | b4 | bx |
|-----|-----|-----|-----|-----|-----|-----|
| Q10 | 1.397 | *-3.357* | -1.928 | -0.596 | 0.940 | -1.235 |
| Q11 | 1.262 | *-3.090* | -1.335 | -0.162 | 1.823 | -0.691 |
| Q30 | 1.725 | *-3.531* | -1.762 | -0.670 | 0.822 | -1.285 |
| Q4 | 1.824 | *-3.071* | -1.729 | -0.595 | 1.320 | -1.019 |
| Q23 | 1.793 | -2.725 | -1.311 | -0.239 | 1.441 | -0.708 |
| Q24 | 2.612 | *-3.050* | -1.417 | -0.416 | 1.110 | -0.943 |
| Q7 | 1.151 | *-3.420* | -1.578 | -0.360 | 1.828 | -0.883 |
| Q26 | 1.461 | -2.592 | -1.074 | -0.027 | 1.791 | -0.475 |
| Q27 | 1.944 | -2.535 | -1.278 | -0.085 | 1.527 | -0.593 |
| M | 1.686 | *-3.041* | -1.490 | -0.350 | 1.400 | -0.870 |
| SD | 0.416 | 0.341 | 0.260 | 0.223 | 0.360 | 0.263 |

M: mean; SD: Standard Deviation.

Regarding item's difficulty parameters ($b1$—$b4$, and $bx$). The difficulty average ($bx$) of all items is within the acceptable range [−3; +3] for the IRT measure. Although certain difficulty parameter values for the item level (**b1**) are less than −3 (as shown in italics), they are all close to this threshold. Based on these results, we conclude that the FSS-BR-S is adequate for measuring the flow state of individuals during typical activities.

Fig 5 depicts the Test Information Curve for evaluating the precision of the FSS-BR-S. This graph demonstrates that our instrument is more accurate (reliable) between the range of [−5; 3]. In other words, the FSS-BR-S provides a more accurate estimation of the flow state of individuals with a theta value ($\theta$) within this range.

## Discussion

In this study, we developed and validated an alternative short version of the FSS-2 containing nine items with the highest IRT-calculated discriminant item values from the long version of the FSS-2. The short version given in this study represents our most significant contribution to the research literature. Its most significant practical implication is that our proposed short version requires less effort and less time to be applied than the large version.

The validation processes of psychometric instruments have two major goals: (1) the first focuses on the appropriation and relevance of original concepts in the new culture and for other domains; and (2) the second focuses on the validation of instruments for cross-cultural studies. Although the former goal was the primary emphasis of our study, we also contributed to the latter goal. This study is also considered part of a secondary study in which several translated versions of FSS-2 will be evaluated in order to have a standard instrument for measuring the flow state in Brazilian-Portuguese-speaking people and for general activities.

A good questionnaire includes a small number of items and a maximum response time of 10 to 15 minutes [48]. We believe, for this pragmatic reason, that the FSS-BR-S should be preferred for measuring flow state. According to the Occam's razor principle (also known as the law of parsimony), the simplest explanation is most often the correct one [49, 50].

In this study, we also identified the discrimination and difficulty parameters for all FSS-BR-S items by IRT analysis. These criteria are measures of the item's quality, and their adequacy indicates a high discriminatory power. The Test Information Curve stated that the latent variable they tested is close to the expected interval, indicating that the instrument accurately evaluates the flow state of the individuals.

## Test Information and Standard Errors

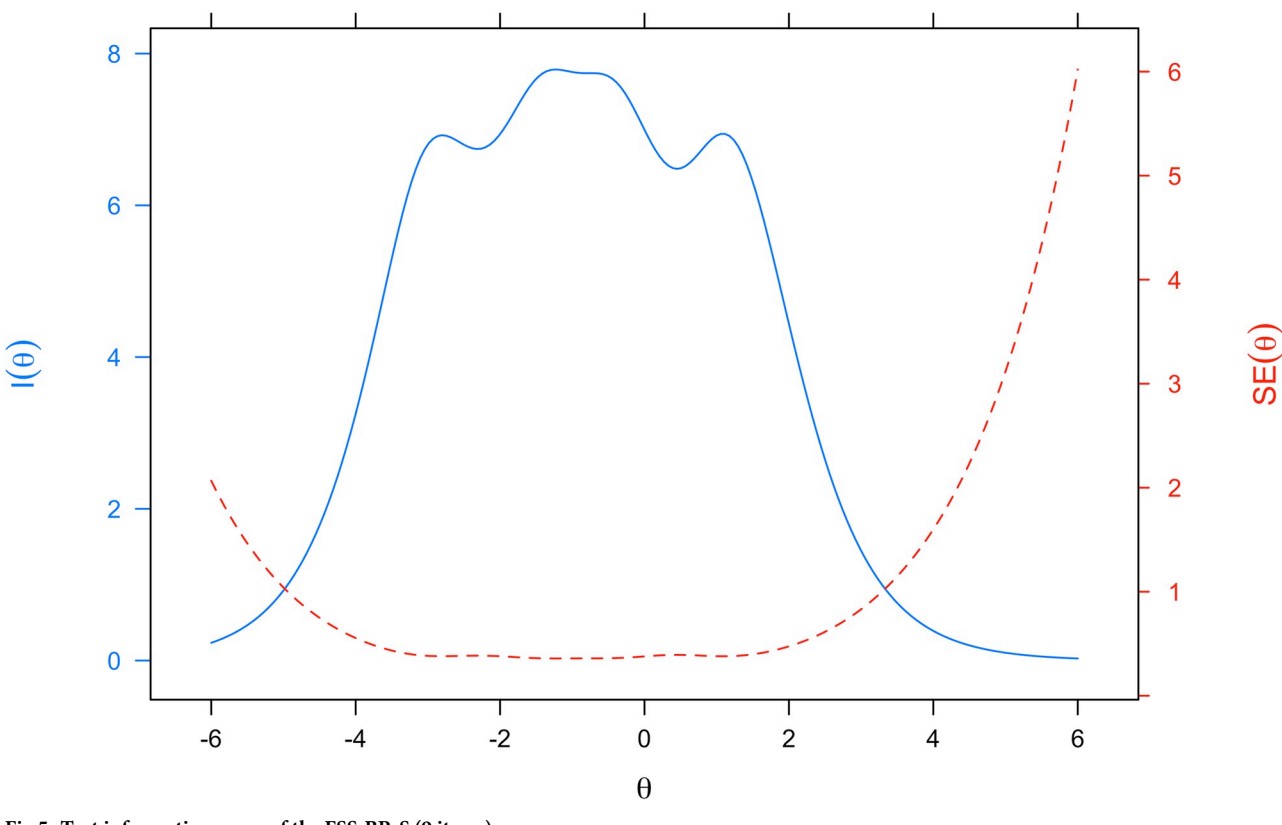

**Fig 5. Test information curve of the FSS-BR-S (9 items).**

## Conclusion and future work

This study presents the validation and psychometric aspects of the FSS-BR-S (instrument for measuring flow state with nine items). According to our findings, the FSS-BR-S is a suitable instrument for measuring flow state in Brazilians and for conducting cross-cultural study.

Our study is one of the first to develop a shorter version of the FSS-2, providing evidence of its validity. This adaption (known as FSS-BR-S) was done and verified as a nine-item questionnaire that has been demonstrated to be adequate for measuring the flow state of Brazilian population, which is another significant practical implication of our study. Our instrument cannot be used for diagnosis, but it may be used in any Brazilian environment and for any activity to obtain a deeper understanding of the flow experience.

The selection of a non-probabilistic and non-representative sample is our study's most significant flaw. However, this fact does not undermine the primary contribution of our study, which is the development and validation of the FSS-BR-S as a suitable instrument for measuring flow status. In future research studies, we should investigate how different groups of Brazilians (divided by gender, age, social status, ethnicity, and activity level) understand the items in our instrument because our current dataset lacks the representation and sample size necessary to do these studies. Therefore, we will continue to collect additional data for MGCF analysis and DIF analysis. We will apply probabilistic sampling with a representative and larger sample

size, comprised of people from the five regions of Brazil of varying age, gender, ethnicity, and socioeconomic position.

Our current study did not examine the criterion validity, also known as the external validity, of a psychometric instrument. Future research will conduct this type of validation to compare the individual flow state scale to other psychometric scales (e.g., anxiety). The purpose of this study will be to validate and compare the association between flow state, anxiety, well-being, happiness, serenity, and satisfaction.

## Data availability

All relevant data are within the manuscript and its Supporting Information files.

We also made available R code used for running our validation available on a GitHub repository at https://github.com/geiser/validation-FSS-BR.

## Supporting information

**S1 Appendix. FSS-BR.** Portuguese-Brazilian Long Version of the Flow-State Scale 2.
(PDF)

**S2 Appendix. FSS-Short.** Portuguese-Brazilian Short Version of the Flow-State Scale 2.
(PDF)

**S1 Dataset. FSS dataset.** Dataset with the responses gathered to validate the Portuguese-Brazilian Flow-State Scale 2.
(CSV)

## Author Contributions

**Conceptualization:** Ig Ibert Bittencourt, Seiji Isotani.

**Data curation:** Geiser Chalco Challco, Alexandre Marinho.

**Formal analysis:** Leogildo Freires, Jorge Coelho, Julio Costa.

**Funding acquisition:** Ig Ibert Bittencourt, Yu Lu, Yang Pian.

**Investigation:** Geiser Chalco Challco.

**Methodology:** Leogildo Freires, Sheyla Fernandes, Seiji Isotani.

**Project administration:** Yu Lu, Yang Pian.

**Resources:** Jorge Coelho, Alexandre Marinho.

**Software:** Geiser Chalco Challco, Jorge Coelho, Julio Costa, Alexandre Marinho.

**Supervision:** Sheyla Fernandes.

**Validation:** Leogildo Freires, Julio Costa.

**Visualization:** Seiji Isotani.

**Writing – original draft:** Ig Ibert Bittencourt, Geiser Chalco Challco.

**Writing – review & editing:** Yu Lu, Geiser Chalco Challco, Yang Pian.

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
