## [Decision Letter · Decision Letter 0]

13 Dec 2022

PONE-D-22-16094Brazilian-Portuguese Flow State Scale 2 (FSS-BR): Psychometric PropertiesPLOS ONE

Dear Dr. Bittencourt,

Thank you for submitting your manuscript to PLOS ONE. After careful consideration, we feel that it has merit but does not fully meet PLOS ONE’s publication criteria as it currently stands. Therefore, we invite you to submit a revised version of the manuscript that addresses the points raised during the review process.Your work is really a valuable and interesting one. After a very careful evaluation of our experts and mine,there are some issues,we would invite you to consider for revision:1. Please ensure a proofreading assistance in writing the standard English. Unfortunately, I found many writing errors that can overwhelm the reader and probably deviate  from the main aim of your very interesting work ;2. Kindly  consider the reviewer 2 comments regarding the validation and reliability  of the presented models figured in Tables 2-4 and 8 regarding and related results;3. Consider to write a paragraph specifying some of the results reached by the semantic assessment of ten people who checked the relevance of the instrument for the targeted population in which you based the utility of FSB for the present work (line 148-154).4. Please pay attention either to all the reviewers' concerns and comments.

We look forward to receiving your revised manuscript.

Kind regards,

Silva Ibrahimi, PhD

Academic Editor

PLOS ONE

Journal Requirements:

a) Did participants provide their written or verbal informed consent to participate in this study?

“This work has been supported by the institutions: Conselho Nacional de 503 Desenvolvimento Cient´ıfico e Tecnol´ogico (CNPq), Coordena¸c˜ao de Aperfei¸coamento 504 de Pessoal de N´ıvel Superior (CAPES), Funda¸c˜ao de Amparo `a Ciˆencia e Tecnologia 505 do Estado de Alagoas (FAPEAL), and Beijing Normal University (BNU)”

“The study was funded by Beijing Advanced Innovation Center for Future Education

(AICFE) and the Conselho Nacional de Desenvolvimento Científico e Tecnológico

(CNPq). Researchers from BNU participated in the data analysis, preparation of the

manuscript and paper review”

Reviewers' comments:

Reviewer's Responses to Questions

**Comments to the Author**

1. Is the manuscript technically sound, and do the data support the conclusions?

Reviewer #1: Yes

Reviewer #2: Partly

Reviewer #3: Partly

2. Has the statistical analysis been performed appropriately and rigorously? 

Reviewer #1: I Don't Know

Reviewer #2: No

Reviewer #3: Yes

3. Have the authors made all data underlying the findings in their manuscript fully available?

Reviewer #1: Yes

Reviewer #2: No

Reviewer #3: Yes

4. Is the manuscript presented in an intelligible fashion and written in standard English?

Reviewer #1: Yes

Reviewer #2: No

Reviewer #3: Yes

5. Review Comments to the Author

Reviewer #1: You did an excellent job describing the measurements you were evaluating for its reliability and validity and the reason why. I was a little concerned that over 45% of your participants were under the age of 18 years because those under 18 may not fully understand the entire instrument given. In the data analysis procedure, it was clear as to why each assessment was used.

In the results section, I had difficulty following what you had found. I felt like a lot of numbers and acronyms were shown but I became overwhelmed with all of the info being given at once and as a result did not understand what it all meant.

For example, you define nine different acronyms (lines 280-283) and then use only the acronyms after that. It is hard for the reader to remember all of the acronyms and process all of the numbers with them. So the reader must go back and forth in the paper to try and recall what each acronym stands for and then what the psychometrics mean for it. I recommend revising this section of the paper so that it is more manageable for the reader to follow everything that is being reported.

In the discussion and conclusion sections, they were clear and was supported by your findings.

Overall, the article was well written and thought out, except for the results section. I believe that this research is important to disseminate but it is just not ready yet.

Reviewer #2: I have read the paper entitled Brazilian-Portuguese Flow State Scale 2 (FSS-BR): Psychometric Properties and the respective review process that has been developed.

I would like to commend the authors for the attention given to the psychometric evaluation of this instrument. I believe this is a noteworthy path for quality research.

I also appreciated the effort taken in addressing some of the reviewers concerns. I felt that their criticism is, in general, acceptable. I found some of the responses given a little sort than expected, but I will leave that consideration for the respective reviewers and editor.

As for my analysis of the work developed, unfortunately, I have found many issues that would take considerable time to express in this review. I will point only two, given that I believe that one of them will require a substantial reorganization of the paper.

During my reading I found several issues with the written English. I can relate to the authors, being a non-native English language researcher. However, some sections are in need of greater attention. This can be seen in the introduction and sample sections. Please make the amendments needed to avoid this type of criticism.

However, my major criticism relates to the quality of the final models obtained (Tables 2, 4, and 8). I disagree with your claim that "analysis indicates that both versions have an adequate psychometric properties". For example, in page 6 the authors present several cutoff values for multiple indices. I disagree with the 0.80 value as acceptable for CIF and TLI, and partially, with SRMR and RMSEA < 0.08 as generally admissible. In fact, when consulting the given reference (e.g., 30; Byrne & Campbell, 1999), I have not found support for these cut-off values. In a more recent and authoritative textbook (which is aligned with several others on this topic), it is presented the following:

"CUTOFF VALUES FOR FIT INDICES: THE MAGIC .90, OR IS THAT .95? Although we know we need to complement the χ2 with additional fit indices, one question still remains no matter what index is chosen: What is the appropriate cutoff value for that index? For most of the incremental fit statistics, accepting models producing values of .90 became standard practice in the early 1990s. However, the case was made that .90 was too low and could lead to false models being accepted,

and by the end of the decade .95 had become the standard for indices such as the TLI and CFI [25]. In general, .95 somehow became the magic number indicating good-fitting models." (Hair et al., 2014, p. 582).

No mention of the 0.80 (CFI/TLI) values is here presented. Moreover, in the same textbook but on page 584, the accepted values of RMSEA and CFI according to the sample size and parameters to be estimated, for your case, would suggest a minimal CFI of .92 and RMSEA of < 0.07. Although several authors indicate that these values should be used as a somewhat flexible rule, this manuscript results are in fact demonstrative of problems with the factorial structure of the instrument, and therefore, I cannot consider it as having adequate psychometric properties.

As another example of psychometric issues, in the FSS-BR alternative short version, the authors have chosen to select items based in their high discriminant value (pag. 12). However, this could be made (although not the perfect approach) by selecting the items with high factorial weight. Your strategy was to select items that are sufficiently different, but as a backlash, you do not know if they evaluate what is intended or if they represent the theoretical factor, only that they are different. For me to evaluate how this impacted the adjustment of the model, the authors must present in table 8 the x2 and dl, and not only the x2/df. Without this, I am inclined to suggest that some serious issues existed in order to change a very bad adjustment (original short) to an excellent adjustment (alternative short).

I hope my comments can help to improve this and further works. Thank you for your effort.

Reviewer #3: Comments

Comment 1. Describe better how the recruitment of participants was carried out. Did the sampling snowball? Did it have seeds? It is not clear how the recruitment was carried out.

Comment 2. As it was a research carried out in a virtual environment, I recommend also mention Circular Letter No. 2 of February 24, 2021, as it is a survey that used the virtual environment for data collection.

Quote the reference:

Brazil. Ministry of Health. National Commission of Ethics and Research. Circular Letter 2 of February 24, 2021. Brasília, 2021.

Comment 3. How was the assessment of the understanding of the instrument carried out by the target population (step carried out with 10 subjects)? Was an instrument used to assess understanding? Was it evaluated using the content validity index?

Comment 4. Table 1 must be inserted in the results section.

Comment 5. The results of the translation stage and evaluation of the instrument's comprehensibility by the target population (stage with 10 subjects) were not described. I recommend briefly describing these results, especially about the understanding assessment stage.

6. PLOS authors have the option to publish the peer review history of their article (what does this mean?). If published, this will include your full peer review and any attached files.

Reviewer #1: No

Reviewer #2: No

Reviewer #3: **Yes: **Noélle O. Freitas

---

## [Author Response · Author response to Decision Letter 0]

26 Jan 2023

Responding to Reviewer Comments

Dear Academic Editor and Reviewers of PLOS ONE, We would like to thank you for your insightful comments and suggestions; each has been carefully analyzed and considered in the new version of the paper. Based on the comments, criticisms, and suggestions, we realized that our main objective was to propose a short version of the FSS-2. In this way, the most current version of the manuscript was adjusted in this direction. We hope that all these changes fulfill the requirements to make the manuscript ready for publication.

General comments:

Comment: 

“The study was funded by Beijing Advanced Innovation Center for Future Education

(AICFE) and the Conselho Nacional de Desenvolvimento Científico e Tecnológico

(CNPq). Researchers from BNU participated in the data analysis, preparation of the

manuscript and paper review”

Answer: 

Dear Reviewer, thank you for your comment. We chose to remove the acknowledgement section because the financing statement would be put in another section. The current funding statement is adequate.

Review comments to the author:

Reviewer 1: You did an excellent job describing the measurements you were evaluating for its reliability and validity and the reason why. I was a little concerned that over 45% of your participants were under the age of 18 years because those under 18 may not fully understand the entire instrument given. In the data analysis procedure, it was clear as to why each assessment was used.

Answer: 

Dear reviewer, thank you for your feedback; nonetheless, in our study, we also aim to show evidence that the Brazilian-Portuguese Flow State Scale 2 (FSS-2) is comprehensible to anyone who can read Portuguese. In this regard, there is no reason to suppose that people under the age of 18 have trouble comprehending the FSS-2, given that 45% of our participants are between the ages of 15 and 18 y/o. All of our respondents under the age of 18 clearly understand the questions and can scale their responses appropriately. Other studies in the literature with individuals under the age of 18 are, for example, the Fournier and colleagues' (2007) study, which included participants ranging from 12 to 68 years old. More information is available at:

Fournier, J., Gaudreau, P., Demontrond-Behr, P., Visioli, J., Forest, J., & Jackson, S. (2007). French translation of the Flow State Scale-2: Factor structure, cross-cultural invariance, and associations with goal attainment. Psychology of Sport and Exercise, 8(6), 897–916. https://doi.org/10.1016/j.psychsport.2006.07.007/

In addition, in the most recent version of our manuscript, in the section "Procedure," we added the following paragraph to emphasize the semantic validation carried out in our study:

We translated and adapted the original English version of Flow State Scale (FSS-2). The translation and adaptation were completed in accordance with the International Test Commission's procedures. Two separate interpreters performed the forward translation of items from the original English version prepared by Martin and Eklund (2002), amended by Jackson, Martin, and Eklund (2008), and published in 2010. A third independent translator then reversed the translation from Brazilian Portuguese to English. This semantic assessment was carried out by ten persons to determine whether the instrument could be comprehended by the people for whom it was designed. This assessment was completed in two stages, with five participants participating in each stage. Participants were initially asked to read the instrument item by item. After reading each item, participants were asked about their understanding of the item in the second stage. When there was a doubt regarding words or sentences, it was recorded and corrections were made, but no significant changes were required.

Reviewer 1 - Comment 1: In the results section, I had difficulty following what you had found. I felt like a lot of numbers and acronyms were shown but I became overwhelmed with all of the info being given at once and as a result did not understand what it all meant.

For example, you define nine different acronyms (lines 280-283) and then use only the acronyms after that. It is hard for the reader to remember all of the acronyms and process all of the numbers with them. So the reader must go back and forth in the paper to try and recall what each acronym stands for and then what the psychometrics mean for it. I recommend revising this section of the paper so that it is more manageable for the reader to follow everything that is being reported.

Answer: 

Dear Reviewer, your appointment was critical in revising the results section's content. We believe that the most recent version of our manuscript is more adequate. We removed many acronyms, and we incorporated legends for all acronyms in all of the tables.

Reviewer 1 - Comment 2: Overall, the article was well written and thought out, except for the results section. I believe that this research is important to disseminate but it is just not ready yet.

Answer: 

Dear Reviewer, your notes were essential to adjusting the content of the Results section. We consider that the most current version of the manuscript is adequate and uses clear and concise writing.

Reviewer 2 - Comment 3: I would like to commend the authors for the attention given to the psychometric evaluation of this instrument. I believe this is a noteworthy path for quality research.

Answer: 

Dear Reviewer, thank you for acknowledging our effort and commitment in conducting the study and writing the manuscript.

Reviewer 2 - Comment 4: As for my analysis of the work developed, unfortunately, I have found many issues that would take considerable time to express in this review. I will point only two, given that I believe that one of them will require a substantial reorganization of the paper.

During my reading I found several issues with the written English. I can relate to the authors, being a non-native English language researcher. However, some sections are in need of greater attention. This can be seen in the introduction and sample sections. Please make the amendments needed to avoid this type of criticism.

Answer: 

Dear Reviewer, your careful reading and observation were considered, and the text of the manuscript was revised. In this way, we consider that the most current version of the manuscript is adequate and uses a more clear and concise writing.

Reviewer 2 - Comment 1: However, my major criticism relates to the quality of the final models obtained (Tables 2, 4, and 8). I disagree with your claim that "analysis indicates that both versions have an adequate psychometric properties". For example, in page 6 the authors present several cutoff values for multiple indices. I disagree with the 0.80 value as acceptable for CIF and TLI, and partially, with SRMR and RMSEA < 0.08 as generally admissible. In fact, when consulting the given reference (e.g., 30; Byrne & Campbell, 1999), I have not found support for these cut-off values. In a more recent and authoritative textbook (which is aligned with several others on this topic), it is presented the following:

"CUTOFF VALUES FOR FIT INDICES: THE MAGIC .90, OR IS THAT .95? Although we know we need to complement the χ2 with additional fit indices, one question still remains no matter what index is chosen: What is the appropriate cutoff value for that index? For most of the incremental fit statistics, accepting models producing values of .90 became standard practice in the early 1990s. However, the case was made that .90 was too low and could lead to false models being accepted, and by the end of the decade .95 had become the standard for indices such as the TLI and CFI [25]. In general, .95 somehow became the magic number indicating good-fitting models." (Hair et al., 2014, p. 582).

No mention of the 0.80 (CFI/TLI) values is here presented. Moreover, in the same textbook but on page 584, the accepted values of RMSEA and CFI according to the sample size and parameters to be estimated, for your case, would suggest a minimal CFI of .92 and RMSEA of < 0.07. Although several authors indicate that these values should be used as a somewhat flexible rule, this manuscript results are in fact demonstrative of problems with the factorial structure of the instrument, and therefore, I cannot consider it as having adequate psychometric properties.

Answer: 

Dear Reviewer, thank you for your careful observations. Your notes were essential to adjusting the content of the Data Analysis section. With the statement "analysis indicates that both versions have adequate psychometric properties," we meant to imply that the findings of our analysis using both models (2nd order and multi-correlation models) were adequate to develop our short version of FSS-2 by selecting one item from each of nine factors. To avoid any confusion, we removed the statement and similar claims. We also update the references about the cutoff values as follows:

Lavaan Package [1] was utilized to perform the CFA without requiring the setting of preset parameters, starting values, modifiers, or error values. The following parameters of the adjustment indices were used to examine the outcomes in order to evaluate the model's quality: 

● Chi-square/degrees of freedom (χ²/df) in which values less than 2 indicated an excellent fit, values between 2 and 3 suggested a good fit, and values up to 5 indicated an acceptable fit [2,3].

● The Comparative Fit Indicator (CFI) is an additional model adjustment index where values equal to or better than 0.95 indicate a valid model [4].

● The Tucker Lewis Index (TLI) is a model modification comparison index where values of 0.95 or higher are judged appropriate [4].

● The Root Mean Square Error of Approximation (RMSEA), which is defined as the difference between the model and the observed data, has values less than or equal to 0.05, indicating a great fit [4].

 References 

1. Rosseel Y. Lavaan: An R package for structural equation modeling and more. Version 0.5–12 (BETA). Journal of statistical software. 2012;48(2):1–36.

2. Byrne BM. Structural equation modeling with Mplus: Basic concepts, applications, and programming. routledge; 2013.

3. Saskia van Laar & Johan Braeken (2022) Caught off Base: A Note on the Interpretation of Incremental Fit Indices, Structural Equation Modeling: A Multidisciplinary Journal, 29:6, 935-943, DOI: 10.1080/10705511.2022.2050730.

4. Carmen Ximénez, Alberto Maydeu-Olivares, Dexin Shi & Javier Revuelta (2022) Assessing Cutoff Values of SEM Fit Indices: Advantages of the Unbiased SRMR Index and Its Cutoff Criterion Based on Communality, Structural Equation Modeling: A Multidisciplinary Journal, 29:3, 368-380, DOI: 10.1080/10705511.2021.1992596.

We also want to emphasize that the primary goal of our research is to create a condensed version of the FSS-2. In this regard, the cutoff value found for our proposed short version was quite excellent χ² = 44.36, p < 0.001, χ²/df = 1.61, CFI = 0.990, TLI = 0.987, RMSEA = 0.039 (0.015 - 0,060).

Reviewer 2 - Comment 2: As another example of psychometric issues, in the FSS-BR alternative short version, the authors have chosen to select items based in their high discriminant value (pag. 12). However, this could be made (although not the perfect approach) by selecting the items with high factorial weight. Your strategy was to select items that are sufficiently different, but as a backlash, you do not know if they evaluate what is intended or if they represent the theoretical factor, only that they are different. For me to evaluate how this impacted the adjustment of the model, the authors must present in table 8 the x2 and dl, and not only the x2/df. Without this, I am inclined to suggest that some serious issues existed in order to change a very bad adjustment (original short) to an excellent adjustment (alternative short).

Answer: 

Dear Reviewer, the chi-square and degree of freedom values have been included to Table 8 to ensure our good faith and transparency. Raw data will also be made available in supplemental material. Given that we are reporting the relevant metrics, we are meeting the requirements.

We also want to make it clear that using the "a" parameter is a legitimate technique. The "a" parameter indicates how well an item differentiates examinee responses. A greater score indicates that the object separates more effectively. Because the purpose of most psychometric measures is to discriminate between low and high values of a scale, items with higher discriminations ("a" parameter) are generally thought to be superior. Items with very low parameters provide little measurement information and may be changed or removed. Furthermore, the technique we used will allow us to conduct future research of criterion validity - the gold standard to validate a psychometric measurement instrument. In this sense, we update the data analysis procedure, including the following texts:

Two models were used to evaluate the structures of the short version of the FSS-BR-S in the CFA: the unidimensional model with the original 9 items published in the manual of Jackson et al. (2010); and an alternative unidimensional model that includes items with greater discrimination power - the ability to differentiate between individuals with a low overall flow state and those with a high overall flow state. In other words, to select items for the alternative unidimensional model of the FSS-BR-S, the discrimination parameter (a) from an Item Response Theory (IRT) analysis was used to provide an index of how well the item distinguishes between high and low flow states. Since the objective of most psychometric measures is to differentiate between examinees' responses, items with a higher discrimination parameter (a) are typically considered to be superior. Items with low parameters provide minimal measurement data and may be evaluated for replacement or elimination [1]. The method we utilized will allow future studies to do criterion validity tests, which are the gold standard for identifying the degree of goodness with which an instrument's results represent its measurement [2].

References:

1. Guyer, R., & Thompson, N.A. (2014). User’s Manual for Xcalibre item response theory calibration software, version 4.2.2 and later. Woodbury MN: Assessment Systems Corporation.

2. Prinsen, C. A., Vohra, S., Rose, M. R., Boers, M., Tugwell, P., Clarke, M., ... & Terwee, C. B. (2016). Guideline for selecting outcome measurement instruments for outcomes included in a Core Outcome Set. The Netherlands: COMET COSMIN.

Reviewer 3 - Comment 1: Describe better how the recruitment of participants was carried out. Did the sampling snowball? Did it have seeds? It is not clear how the recruitment was carried out.

Answer: 

Dear Reviewer, your input was invaluable in enhancing the content of the Recruitment section. We believe that the most recent version of our manuscript resolves your concern. This new section is as follows:

The data gathering procedure was completely carried out via the Internet, with respondents' voluntary participation. To acquire responses to the questionnaires used in this study, researchers distributed recruiting messages via their own social media networks (e.g., Facebook, Instagram, Whatsapp) and e-mails. There was no specific group targeted on social media networks. 

The invitation to participate in the survey did not employ lists that allow third parties to identify the visitors or visualize their contact data (e-mail, phone, etc.). After completing various activities, the participants completed the Brazilian Portuguese-translated version of the items corresponding to the FSS-2 questionnaire, which was made available in electronic form. When the questionnaire was halted, respondents might restart it later as long as they did not remove their browsing history. The form was open for 180 days, during which time we collected the data used in this study. During this time, the researchers involved were also available to answer any questions the participants had about the study.

Reviewer 3 - Comment 2: As it was a research carried out in a virtual environment, I recommend also mention Circular Letter No. 2 of February 24, 2021, as it is a survey that used the virtual environment for data collection.

Quote the reference:

Brazil. Ministry of Health. National Commission of Ethics and Research. Circular Letter 2 of February 24, 2021. Brasília, 2021.

Answer: 

Dear Reviewer, thank you. Your consideration is very important. It is important to note that the research project was reviewed and approved by the Human Research Ethics Committee of the Federal University of Alagoas (UFAL) (with Protocol No. 35701820.3.0000.5013), while also following the guidelines in Circular Letter No. 2/2021/CONEP/SECNS/MS. We believe that the most recent version of the manuscript takes your comments into account. The following is the new Ethics Approval section: 

We strictly adhered to the Brazilian National Health Council resolutions 466/12 and 510/16. (CNS). Furthermore, we strictly adhered to all of the standards outlined in Circular Letter No. 2/2021/CONEP/SECNS/MS. This research investigation was approved by the Human Research Ethics Committee of the Federal University of Alagoas (UFAL) (Protocol No. 35701820.3.0000.5013). As a result, the participants were informed that they were not required to participate in the research and that they might withdraw at any time if they were uncomfortable for any reason. Prior to answering the questionnaire, the participants (and their parents in the case of minors) agreed to a Free Prior and Informed Consent (FPIC) in which we informed the participants that their provided information would be confidential, with no way of individual identification, and that their responses would be analyzed as a whole rather than individually. Each participant spent around twenty ($20$) minutes reading the FPIC and answering the questionnaire.

 Reviewer 3 - Comment 3: How was the assessment of the understanding of the instrument carried out by the target population (step carried out with 10 subjects)? Was an instrument used to assess understanding? Was it evaluated using the content validity index?

Answer: 

Dear Reviewer, thank you for your careful observations. Your notes were critical in revising the Procedure section's content. We believe that the most recent version of the manuscript takes this into account. In the Procedure section, we included the following paragraph to explicitly describe the assessment of understanding of our translated questions.

We translated and adapted the original English version of Flow State Scale (FSS-2). The translation and adaption were completed in accordance with the International Test Commission's procedures. Two separate interpreters performed the forward translation of items from the original English version prepared by Martin and Eklund (2002), amended by Jackson, Martin, and Eklund (2008), and published in 2010. A third independent translator then reversed the translation from Brazilian Portuguese to English. This semantic assessment was carried out by ten persons to determine whether the instrument could be comprehended by the people for whom it was designed. This assessment was completed in two stages, with five participants participating in each stage. Participants were initially asked to read the instrument item by item. After reading each item, participants were asked about their understanding of the item in the second stage. When there was a doubt regarding words or sentences, it was recorded and corrections were made, but no significant changes were required.

Reviewer 3 - Comment 4: Table 1 must be inserted in the results section.

Answer: 

Dear Reviewer, Thank you for taking the time to read and your comment. We moved Table 1 for the Result Section in the most recent version of our manuscript, following your recommendation.

Reviewer 3 - Comment 5: The results of the translation stage and evaluation of the instrument's comprehensibility by the target population (stage with 10 subjects) were not described. I recommend briefly describing these results, especially about the understanding assessment stage.

Answer: 

Dear Reviewer, Your observations were essential to adjusting the content of the Procedure section. In the Procedure section, we included the following paragraph to explicitly describe how we performed the assessment of comprehensibility of ten people in our translated questions. 

We translated and adapted the original English version of Flow State Scale (FSS-2). The translation and adaption were completed in accordance with the International Test Commission's procedures. Two separate interpreters performed the forward translation of items from the original English version prepared by Martin and Eklund (2002), amended by Jackson, Martin, and Eklund (2008), and published in 2010. A third independent translator then reversed the translation from Brazilian Portuguese to English. This semantic assessment was carried out by ten persons to determine whether the instrument could be comprehended by the people for whom it was designed. This assessment was completed in two stages, with five participants participating in each stage. Participants were initially asked to read the instrument item by item. After reading each item, participants were asked about their understanding of the item in the second stage. When there was a doubt regarding words or sentences, it was recorded and corrections were made, but no significant changes were required.

---

## [Decision Letter · Decision Letter 1]

24 Apr 2023

PONE-D-22-16094R1Psychometric properties of the Brazilian-Portuguese Flow State Scale Short (FSS-BR-S)PLOS ONE

Dear Dr. Bittencourt,

Thank you for submitting your manuscript to PLOS ONE. After careful consideration, we feel that it has merit but does not fully meet PLOS ONE’s publication criteria as it currently stands. Therefore, we invite you to submit a revised version of the manuscript that addresses the points raised during the review process.

ACADEMIC EDITOR: Dear authors,Many thanks for your revisions on your manuscript. As you will see from the reviews below, both reviewers are satisfied with the revisions. I am happy to inform you that, after you will make the small changes suggested by Reviewer #1, the manuscript can be accepted fro publication.I look forward to receiving a final version of the manuscript.

We look forward to receiving your revised manuscript.

Kind regards,

Nicola Diviani

Academic Editor

PLOS ONE

Journal Requirements:

Reviewers' comments:

Reviewer's Responses to Questions

**Comments to the Author**

1. If the authors have adequately addressed your comments raised in a previous round of review and you feel that this manuscript is now acceptable for publication, you may indicate that here to bypass the “Comments to the Author” section, enter your conflict of interest statement in the “Confidential to Editor” section, and submit your "Accept" recommendation.

Reviewer #1: All comments have been addressed

Reviewer #3: All comments have been addressed

2. Is the manuscript technically sound, and do the data support the conclusions?

Reviewer #1: Yes

Reviewer #3: Yes

3. Has the statistical analysis been performed appropriately and rigorously? 

Reviewer #1: I Don't Know

Reviewer #3: Yes

4. Have the authors made all data underlying the findings in their manuscript fully available?

Reviewer #1: Yes

Reviewer #3: Yes

5. Is the manuscript presented in an intelligible fashion and written in standard English?

Reviewer #1: Yes

Reviewer #3: Yes

6. Review Comments to the Author

Reviewer #1: I would like to thank the authors for making all of the recommended revisions. The manuscript is much clearer and is easier to read.

Some minor revisions remain, mostly typos. They are as follows:

line 165--change "constrct" to "construct"

line 168--change "TRI" to "IRT"

line 177--move "(CFA)" to line 176 after the word "Analysis"

line 336--Is the range supposed to be -3, +3? Currently it states 3, +3

line 386--changes "quetions" to "questions"

line 411--change "rang" to "range"

line 456 change "this" to "these"

Once these minor revisions are made, I recommend this article for publication.

Reviewer #3: (No Response)

7. PLOS authors have the option to publish the peer review history of their article (what does this mean?). If published, this will include your full peer review and any attached files.

Reviewer #1: **Yes: **Jill A. Yamashita

Reviewer #3: **Yes: **Noélle de Oliveira Freitas

---

## [Author Response · Author response to Decision Letter 1]

17 May 2023

Responding to Reviewer Comments

Dear Academic Editor and Reviewers of PLOS ONE, we would like to thank you for your comments; each has been carefully analyzed and considered in the new version of the paper. The most current version of the manuscript was adjusted in this direction and te hope that all these changes fulfill the requirements to make the manuscript ready for publication.

Review comments to the author:

Reviewer 1 - Comment 6: I would like to thank the authors for making all of the recommended revisions. The manuscript is much clearer and is easier to read.

Some minor revisions remain, mostly typos. They are as follows:

line 165--change "constrct" to "construct"

line 168--change "TRI" to "IRT"

line 177--move "(CFA)" to line 176 after the word "Analysis"

line 336--Is the range supposed to be -3, +3? Currently it states 3, +3

line 386--changes "quetions" to "questions"

line 411--change "rang" to "range"

line 456 change "this" to "these"

Once these minor revisions are made, I recommend this article for publication.

Answer: 

Dear Reviewer, thanks for your observation and comments. The text of the manuscript was revised, and we removed all typos.

---

## [Editor Report · Decision Letter 2]

22 May 2023

Psychometric properties of the Brazilian-Portuguese Flow State Scale Short (FSS-BR-S)

PONE-D-22-16094R2

Dear Dr. Bittencourt,

We’re pleased to inform you that your manuscript has been judged scientifically suitable for publication and will be formally accepted for publication once it meets all outstanding technical requirements.

Kind regards,

Nicola Diviani

Academic Editor

PLOS ONE
---

## [Editor Report · Acceptance letter]

20 Jan 2024

PONE-D-22-16094R2 

PLOS ONE

Dear Dr. Bittencourt, 

I'm pleased to inform you that your manuscript has been deemed suitable for publication in PLOS ONE. Congratulations! Your manuscript is now being handed over to our production team.

Kind regards, 

on behalf of

Dr. Nicola Diviani 

Academic Editor

PLOS ONE